# Research

ecology, bioinformatics, microbiology

microbiome, functional trait, niche partitioning, herbivory, coral reef

**Authors for correspondence:**
Jarrod J. Scott
e-mail: jarrod.jude.scott@gmail.com
Douglas B. Rasher
e-mail: drasher@bigelow.org

# Intestinal microbes: an axis of functional diversity among large marine consumers

Jarrod J. Scott[1], Thomas C. Adam[2], Alain Duran[4], Deron E. Burkepile[2,3] and Douglas B. Rasher[5]

[1]Smithsonian Tropical Research Institute, Balboa, República de Panamá
[2]Marine Science Institute, and [3]Department of Ecology, Evolution, and Marine Biology, University of California, Santa Barbara, CA 93106, USA
[4]Department of Biological Sciences, Florida International University, Miami, FL 33199, USA
[5]Bigelow Laboratory for Ocean Sciences, East Boothbay, ME 04544, USA

 JJS, 0000-0001-9863-1318; TCA, 0000-0001-6146-0260; AD, 0000-0002-7847-6426; DEB, 0000-0002-0427-0484; DBR, 0000-0002-0212-8070

Microbes are ubiquitous throughout the world's oceans, yet the manner and extent of their influence on the ecology and evolution of large, mobile fauna remains poorly understood. Here, we establish the intestinal microbiome as a hidden, and potentially important, 'functional trait' of tropical herbivorous fishes—a group of large consumers critical to coral reef resilience. Using field observations, we demonstrate that five common Caribbean fish species display marked differences in where they feed and what they feed on. However, in addition to space use and feeding behaviour—two commonly measured functional traits—we find that interspecific trait differences are even more pronounced when considering the herbivore intestinal microbiome. Microbiome composition was highly species specific. Phylogenetic comparison of the dominant microbiome members to all known microbial taxa suggest that microbiomes are comprised of putative environmental generalists, animal-associates and fish specialists (resident symbionts), the latter of which mapped onto host phylogeny. These putative symbionts are most similar to—among all known microbes—those that occupy the intestines of ecologically and evolutionarily related herbivorous fishes in more distant ocean basins. Our findings therefore suggest that the intestinal microbiome may be an important functional trait among these large-bodied consumers.

## 1. Introduction

Herbivory strongly influences the structure and function of almost all aquatic, shallow marine and terrestrial ecosystems [1–3]. On coral reefs, high rates of herbivory favour corals—the ecosystem's foundation—by excluding upright seaweeds and thick filamentous algae [4,5] that otherwise negatively impact coral reproduction, recruitment, growth and survival [6–8]. Intense herbivory thus increases the recovery potential of corals following disturbance and is becoming increasingly important, given the growing frequency of disturbances to tropical reefs [9]. High rates of herbivory collectively stem from a diverse portfolio of species (especially parrotfishes, surgeonfishes and rabbitfishes) [10,11], because each species feeds on a distinct algal assemblage [12–14]. Consequently, the ability of any given alga to escape attack declines as herbivore species richness increases [13]. Understanding what organismal traits generate, and best capture, this 'complementarity effect' is essential for predicting the impacts of biodiversity loss [15–17] not only on coral reefs [18–20], but in any place where large consumers influence ecosystem structure and function [21,22].

Early research on the functional traits of herbivorous fish centred around jaw morphology and how it may constrain the breadth and type of prey that each fish species is capable of consuming [23–25]. Later, it became apparent that intestinal physiology plays a critical role in dictating what these herbivores actually eat, as it

defines their capacity to acquire nutrients from, and tolerate chemical deterrents produced by, their prey [26–29]. Intestinal physiology may therefore be an important organismal trait upon which functional diversity manifests. More recently, studies of herbivore foraging across different spatial scales [30,31] and gradients of predation risk [32–34] revealed herbivorous fish behaviour (space use) as an additional trait axis upon which functional differences arise. Over the years, such traits have collectively been used to infer niche partitioning among species [35] as well as the potential consequences of biodiversity loss in nature [18,19].

However, despite the strong influence of intestinal physiology on the foraging behaviours of herbivorous fishes and the known role of microbes in vertebrate digestion [36,37], the intestinal microbiomes of herbivorous reef fishes have received little attention compared to terrestrial herbivores [38–41]. Consequently, it remains unknown (i) which fish intestinal microbes are transients (e.g. environmental microbes) versus residents (e.g. symbionts that aid in host digestion) and (ii) whether microbiome composition reflects the functional diversification that has unfolded among these herbivores over evolutionary time [35]. Such unknowns make it impossible to infer which components of the intestinal microbiome are a partial cause versus a consequence of niche partitioning, or whether the collective microbiome reliably serves as a 'functional trait' (sensu [18,19]).

A critical next step for understanding the intestinal microbiomes of herbivorous fishes and their utility as a functional trait is therefore to: (i) decipher the host specificity of each dominant microbiome member, (ii) assess the degree to which microbiome composition reflects fish ecology and/or phylogeny, and (iii) compare microbiomes across a variety of co-occurring fish species that differ markedly in their feeding behaviours. Here, we take these next steps by combining field observations, intestinal microbe metabarcoding, bioinformatics and phylogenetic analyses of five common herbivorous fishes that are critical to the ecology of Caribbean coral reefs.

## 2. Material and methods

### (a) Quantifying herbivore feeding behaviour in nature

To define the foraging behaviours of each herbivorous fish in this study—*Acanthurus coeruleus*, *Acanthurus tractus*, *Scarus taeniopterus*, *Sparisoma aurofrenatum* and *Sparisoma viride*—we characterized individual bites taken by each species at three study sites in the Upper Florida Keys, USA (Conch, French, Molasses reefs), during the boreal summers of 2014 (parrotfishes) and 2016 (surgeonfishes). French and Molasses reefs are well-developed spur-and-groove formations with high architectural complexity while Conch reef is less complex. At each site, we haphazardly selected focal fish over a wide range of sizes and then randomly selected a single bite by each individual to describe (see electronic supplementary material, table S1 for sample sizes). For each bite, we identified the item(s) ingested as well as the characteristics of the target substrate (e.g. hard bottom versus other common substrates such as sponges, gorgonians, etc.) at the location of the bite. For hard substrates, we recorded whether a bite was on a convex, concave or flat surface, and whether that surface was oriented horizontally (less than 45°) or vertically (greater than 45°). We framed each bite with a 5 × 5 cm micro-quadrat and measured the depth of the sediment and height of the algae at several points to determine the average sediment depth and algal height within the immediate vicinity of the bite [42]. We then manually removed sediments and determined whether the fish left a distinct grazing scar

(i.e. calcium carbonate had been removed from the reef framework in addition to epilithic algae).

We visualized the multivariate patterns of herbivory using non-metric multidimensional scaling (NMDS), computed with the metaMDS function from the vegan package [43] within the R environment [44]. For each fish species at each site, we calculated the proportion of bites focused on each prey item (prey variables) as well as the proportion of bites targeting substrates with different characteristics. We also calculated the proportion of bites resulting in a grazing scar. For bites on turf assemblages, we calculated the mean turf height and sediment depth directly adjacent to each bite. Quantitative variables (e.g. sediment depth and turf height) were first rescaled to the range of 0–1. In addition, quantitative and categorical variables were rescaled (so that they would have a similar influence to the 'prey variables') by dividing each variable by the number of prey categories. Rescaled data were then analysed via NMDS using a Bray–Curtis dissimilarity matrix.

### (b) Intestinal sample collection, DNA extraction and sequencing

We collected fish (*A. coeruleus*, $n = 8$; *A. tractus*, $n = 9$; *Sc. taeniopterus*, $n = 9$; *Sp. aurofrenatum*, $n = 13$; *Sp. viride*, $n = 11$) in July 2016 at Pickles reef (25°00′05″ N, 80°24′55″ W), Upper Florida Keys, USA. Pickles reef is a spur-and-groove formation ranging from 5 to 8 m depth, which is intermediate in architectural complexity compared to the reefs where behavioural observations were conducted (Pickles is lower relief than Molasses and French but higher relief than Conch). All four reefs have similar benthic assemblages with low coral cover and high cover of turf algae and macroalgae (particularly *Dictyota* spp.) [45]. Previous work has shown that these herbivore species partition resources similarly among sites [42]. Thus, that behavioural observations were conducted at nearby reefs is unlikely to have affected our ability to link fish microbiomes with differences in foraging ecology. We captured fish using barrier and hand nets (small individuals) or spearing (large individuals). Specimens were immediately returned to the boat and put on ice. Upon return to land (approx. 2 h later), the fish were measured, weighed and dissected using sterile techniques to excise the intestines. Gut segments (fore, mid, hind) were separated, stored in 95% ethanol and frozen until extraction. In brief, we targeted the V4–V5 hypervariable region of the 16S rRNA gene and inferred ASVs. ASVs are analogous to OTUs but with single nucleotide resolution. See the electronic supplementary material for complete details on DNA extraction, sequencing and read processing.

### (c) Community diversity estimates

Microbiome analyses and visualizations were conducted primarily in the R environment [44] using the phyloseq package [46]. Prior to analysis, reads taxonomically classified as mitochondria or chloroplast were removed while retaining all non-chloroplast Cyanobacteria. α-Diversity was calculated using several diversity indices. Shapiro–Wilk normality tests [44,47] indicated that only the Shannon index data were normally distributed. To test whether the Shannon diversity of microbiomes differed among fish species, we employed an analysis of variance (ANOVA) followed by Tukey HSD *post hoc* tests [44,47]. We also tested the non-normal indices using the Kruskal–Wallis test (see electronic supplementary material for details). Microbiome β-diversity was estimated using a Jensen–Shannon divergence index and ordinated with NMDS within the phyloseq package [46]. Differences in microbiome composition among fishes were examined with an analysis of similarity (ANOSIM) in the vegan package [43].

### (d) Inferring microbe habitat (host) specificity

To infer the host specificity of intestinal microbes, we employed an approach modelled after the work of Sullam *et al.* [48]; they

compared fish intestinal microbial datasets to the nr/nt database using BLASTn, and then used phylogenetic inference against top BLASTn hits to classify the habitat preference of intestinal microbes. From phylogenetic clustering, the authors created 14 categories of fish intestinal-associated microbes based on the isolation source of their closest relatives (table 3 in [48]).

Since we were interested in the amplicon sequence variants (ASVs) that statistically defined the microbiomes of each herbivore, we began by first identifying differentially abundant (hereafter 'DA') ASVs across the five fish species using linear discriminant analysis (LDA) effect size (LEfSe) [49] through the Microbiome-Analyst [50] web server. We first filtered the data by setting the minimum ASV read count to 20 and prevalence to 20%, which removed 3796 of the original 11 144 ASVs. The data were then normalized using total sum scaling. We considered only those ASVs with an LDA score greater than 4 and an adjusted $p$-value cut-off of $1.0 \times 10^{-4}$ as DA features, because we found that these more stringent cut-offs eliminated ASVs that were highly abundant in only one or a few individuals of a species and were not consistently abundant across that species. This resulted in 59 DA ASVs.

We used BLASTn against the nr/nt database to identify the closest relatives of each DA ASV and their isolation source. We also screened these ASVs against the Silva Alignment, Classification, and Tree Service (ACT) [51] setting the 'minimum identity with query sequence' to 0.95 and the 'number of neighbours to query sequence' to 5. Any top hits from the BLASTn output not present in the ACT analysis (and vice versa) were added to the fasta file prior to phylogenetic analysis. Similarity searches yielded 297 distinct top hits. To compute the phylogenetic tree, we aligned the 59 DA ASVs and the 297 top hits using mothur [52] against the silva.nr_v132 reference database. We then used a hard mask to trim all reads to the same length (373 bp). Next, we used RAxML [53] and the GTR model for tree computation and the GAMMA rate model for likelihoods. The tree and associated metadata were visualized and annotated using the interactive Tree of Life (iTOL) [54] web server.

We also conducted similarity queries of each DA ASV against the Integrated Microbial Next Generation Sequencing (IMNGS) database [55]—a curated database of short-read sequences scraped monthly from the International Nucleotide Sequence Database Collaboration. A hit to an IMNGS sample was scored positive if the sequence similarity was greater than or equal to 97% and accounted for greater than or equal to 0.1% of total reads from a sample (minimum size for each query set at 200 bp). The number of hits was included as metadata for each ASV in the tree. Similar to Sullam, we then used associated metadata to categorize each top hit into a distinct habitat class (based on its isolation source); however, we tailored the categories to reflect the aims of our study. The isolation source of the closest relative was then used to infer the host specificity of each DA ASV (see Results).

### (e) Associations between intestinal microbes, host phylogeny and herbivore foraging behaviour

We used a series of simple and partial Mantel tests to assess whether DA ASVs were associated with a fish species' foraging ecology and/or phylogenetic history. The dissimilarity matrix for foraging ecology was based on the behavioural foraging data collected on the reef (see above). The phylogenetic dissimilarity matrix was based on a phylogenetic tree we constructed using cytochrome oxidase subunit 1 (COI) genes retrieved from NCBI's Nucleotide Database for the five fish species used in this study with members of the Gerridae used as the outgroup. See the electronic supplementary material for complete details on tree construction. The tree was transformed into a distance matrix using the cophenetic function in R [44]. We then constructed separate dissimilarity matrices to focus on putative resident ASVs

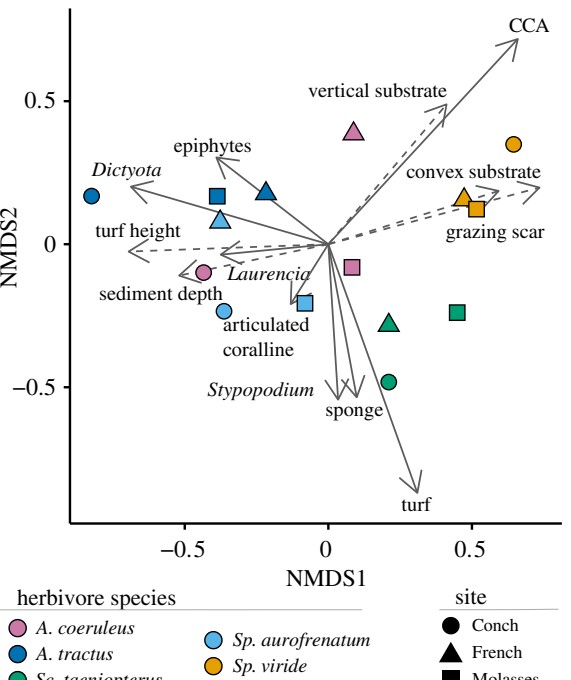

**Figure 1.** Herbivore feeding behaviours. NMDS plot showing the characterization of individual bites made by each species during field observations. Points are mean values for each species at each of three sites in the Florida Keys National Marine Sanctuary (see electronic supplementary material, table S1, for sample sizes). Vector overlays represent the relative influence of the different bite variables on the ordination. Vectors are displayed to distinguish prey targets (solid lines) from associated substrate characteristics (dashed lines). For clarity, only the 13 most influential variables are shown. (Online version in colour.)

and putative environmental ASVs, respectively. All matrices were constructed based on Bray–Curtis dissimilarity of Hellinger transformed data using the vegan package [43] in R [44].

## 3. Results

### (a) Herbivore feeding behaviour

We found that each herbivore species fed from a distinct component of the algal assemblage, with respect to both what they fed on and where they fed (figure 1). NMDS analysis of bite data (electronic supplementary material, table S2) showed that the parrotfish *Sp. viride* fed mostly on sparse, short turf assemblages growing on crustose coralline algae and fed disproportionately from vertical, convex surfaces with few sediments. Most bites by *Sp. viride* left a deep grazing scar where reef carbonate had been excavated. The other species fed most often from flat horizontal surfaces but differed in the types of algae they targeted. The parrotfish *Sc. taeniopterus* fed mostly on short, well-cropped turf assemblages, which it often scraped from the reef resulting in a shallow grazing scar. The parrotfish *Sp. aurofrenatum* showed more diverse behaviours, ingesting articulated coralline algae such as *Amphiroa* spp., other red macroalgae including *Galaxaura* spp., the brown macroalga *Dictyota* spp. and long, sediment-laden algal turfs. Bites by *Sp. aurofrenatum* also occasionally resulted in the removal of reef carbonate, but less often than did bites by *Sc. taeniopterus*. The surgeonfish *A. tractus* disproportionately targeted *Dictyota* spp., and filamentous epiphytes growing on macroalgae and benthic invertebrates. The surgeonfish *A. coeruleus* varied widely among sites in the prey

that it targeted, with individuals feeding on a combination of epiphytes, filamentous turf algae and macroalgae such as *Dictyota*. In contrast with the parrotfishes, bites by *A. tractus* and *A. coeruleus* never left a visible grazing scar on the substrate. Thus, while there is known ontogenetic and site-by-site variation in feeding behaviour [42,45], this variation is small compared to the consistent differences in feeding behaviour we see among these five fish species. Hence, these species exhibit distinct functional trait differences with respect to their morphology and the substrates they target.

## (b) Diversity and composition of the intestinal microbiome

The processed and curated 16S rRNA gene dataset contained approximately 3.12 million high-quality sequences (approx. 375 bp read length). After removing reads classified as mitochondria (35 941 reads) or chloroplast (149 274 reads), the dataset had 2.94 million reads with a range of 11 686–180 159 reads per sample (mean 58 699) (electronic supplementary material, table S3). Modelling and correcting amplicon errors inferred 11 144 ASVs, 63% of which were singleton or doubleton ASVs. The collective herbivore microbiome was largely comprised of ASVs classified as Proteobacteria (Alpha, Delta, Gamma), Firmicutes (Clostrida, Erysipelotrichi), Bacteroidetes (Bacteroidia, Flavobacteriia), Fusobacteria and Planctomycetia (electronic supplementary material, table S4).

However, the relative abundance of reads of each microbial taxonomic group differed markedly among fish species (figure 2a; electronic supplementary material, figure S1), as did the overall diversity of each microbiome (figure 2b; electronic supplementary material, table S3). Specifically, there was a significant difference in microbiome diversity among the five herbivores (ANOVA: $F_{4,45} = 3.91$, $p = 0.008$), with *Sp. aurofrenatum* harbouring a less diverse intestinal microbiome than *Sc. taeniopterus* or *A. tractus* (Tukey's HSD: $p = 0.042$ and 0.010, respectively). Furthermore, when we examined the compositional similarity (i.e. β-diversity) of each microbiome, we found that each herbivore species' intestinal microbiome was unique and non-overlapping (figure 2c). Indeed, ANOSIM revealed herbivore microbiome composition to be highly species specific ($p < 0.001$; $R^2 = 0.85$).

## (c) Defining features of the intestinal microbiome

To identify the microbes that defined each herbivore's intestinal microbiome, we determined which ASVs were differentially abundant (DA) using LDA LEfSe analysis (threshold: 4.0, adjusted $p$-value cut-off: $1.0 \times 10^{-4}$), which revealed 59 DA ASVs across the five fish species (electronic supplementary material, table S5). Cyanobacteria are prey to parrotfish [56] and made up roughly 2% of the total dataset; however, we did not identify any DA ASVs from this group in any herbivore species and thus none appeared in subsequent analyses. Analysis of top BLAST hits from NCBI's nr nucleotide database and phylogenetic comparison to these 59 ASVs revealed the DA ASVs' closest relatives to be either: (i) microbes found broadly throughout the marine environment; (ii) microbes specifically associated with herbivore prey (coral–sponge–algae); (iii) microbes closely associated with animals; or (iv) microbes known only from the intestinal tract of marine herbivorous reef fishes (electronic supplementary material, table S6). We also compared these 59 ASVs to the IMNGS database and found similar patterns

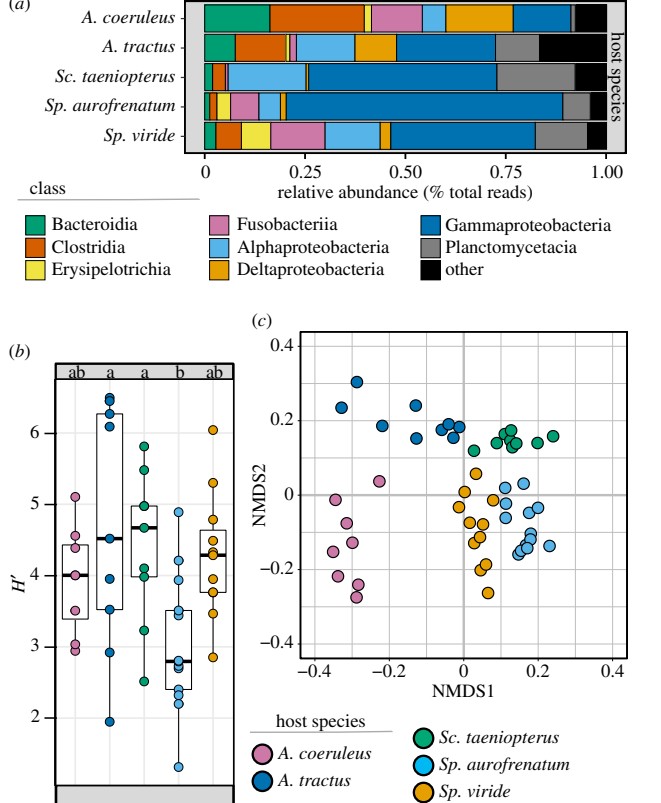

**Figure 2.** Diversity and similarity of intestinal microbiomes. (*a*) Relative read abundance of the dominant bacterial classes found within the intestines of each herbivore. For intraspecific variance see electronic supplementary material, figure S1. (*b*) Microbiome α-diversity. Host species sharing the same letter are not significantly different from each other (Tukey's HSD: $p < 0.5$). (*c*) NMDS ordination of compositional similarity (β-diversity) of each microbiome estimated using the Jensen–Shannon divergence index. (Online version in colour.)

including 13 ASVs that returned no hits to any IMNGS dataset. All ASVs returning no hits against IMNGS were most similar to hits from herbivorous reef fishes found in the NCBI nr and/or Silva database (electronic supplementary material, table S6).

We used phylogenetic inference, BLAST similarity and prevalence in the IMNGS database to classify DA ASVs into the following putative groups: (i) fish specialists—ASVs most closely related to microbes known only from the intestinal tract of herbivorous marine fishes, (ii) animal specialists—ASVs most closely related to microbes from other animals (predominantly intestinal), or (iii) environmental generalists—ASVs most closely related to microbes found broadly in the environment. By and large, the generalist microbes were closely related to microbes isolated from marine or marine-like (e.g. hypersaline mats, saline lakes) habitats including sediments, water, and potential prey (algae, corals, sponges) (figures 3 and 4, table 1; electronic supplementary material, table S6). These were not quantitative—but instead user-guided—determinations, as no appropriate quantitative tools currently exist for assessing microbial habitat preference.

The 33 ASVs we categorized as fish specialists (hereafter 'putative resident symbionts') exhibited the following characteristics: closest database matches and clustering with sequences from the intestinal tracts of other herbivorous marine reef fishes; no 100% matches to the nr database; a low number of hits to the IMNGS database (including all 13 ASVs with zero hits); and taxonomic assignment restricted to groups like the Firmicutes, Desulfovibrionaceae and

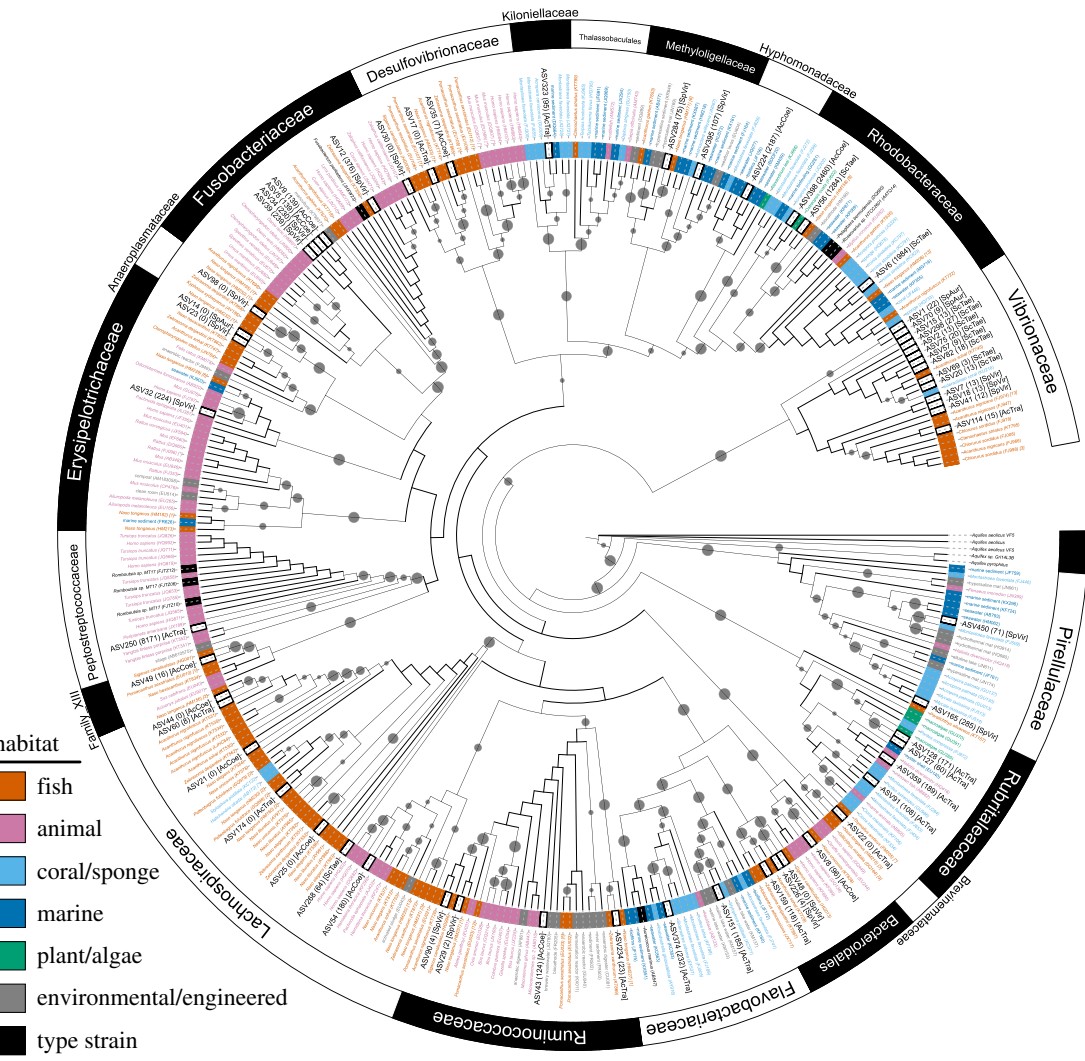

**Figure 3.** Taxonomy and putative origin of the intestinal microbiome. Maximum-likelihood analysis of the 59 DA ASVs and their 297 closest relative sequences. Inner coloured ring corresponds to the assigned habitat category of closest relatives (based on isolation source). Tree leaves for top hits are labelled by their isolation source, a unique code in parentheses corresponding to an accession number, and—where also studied by Sullam et al. [48]—the 'lifestyle' category assigned by those authors in brackets. DA ASVs are labelled by their unique name, by the number of hits to the IMNGS database in parentheses, and in brackets by the fish species in which the ASV was DA. Outer ring depicts family level taxonomic affiliation. Grey dots are bootstrap values greater than 50 and dot size is proportional to the bootstrap value. A higher-resolution and interactive version of this tree—including all annotation data—is available at http://bit.ly/2QA5iTa. (Online version in colour.)

Vibrionaceae (table 1; electronic supplementary material, table S6). By contrast, the 15 ASVs classified as putative environmental generalists (hereafter 'environmental generalists') exhibited the following characteristics: closest database matches and clustering with sequences from sediments, water, corals, sponges and algae; numerous 100% matches to the nr database; a large number of hits to the IMNGS database; and taxonomic assignment restricted to the Alphaproteobacteria, Planctomycetes, Verrucomicrobia and Flavobacteriaceae (figures 3 and 4, table 1; electronic supplementary material, table S6). The host-specific, non-overlapping nature of the herbivore intestinal microbiome (figure 2c) was particularly evident when focused on those microbial taxa that appear to be putative resident symbionts (figure 4).

## (d) Mapping the intestinal microbiome onto herbivore ecology and phylogeny

Putative resident symbionts were strongly correlated with herbivore phylogeny (Mantel $r = 0.845$, $p = 0.008$) but not with feeding behaviour (Mantel $r = 0.251$, $p = 0.259$), indicating that phylogenetically related fishes have similar putative

resident symbionts, but fishes with similar feeding behaviours do not necessarily harbour similar symbionts (figure 4a). For example, only the two surgeonfish species harboured an abundance of Lachnospiraceae (figure 4a), with Acanthurus spp. harbouring six of seven DA ASVs and accounting for 92% of reads across all 221 Lachnospiraceae ASVs. Similarly, only the two species of Sparisoma parrotfish harboured an abundance of Erysipelotrichaceae; they accounted for 78% of all reads from this family (54 total ASVs). Among the Vibrionaceae—the most dominant family in the dataset—13 of the 14 DA ASVs were restricted to the parrotfishes only (figure 4a; electronic supplementary material, table S6).

We even found notable differences among herbivore species in the same genus regarding the putative symbionts they harbour. For instance, A. tractus (but not A. coeruleus) contained the single DA spirochaete (figure 4a) and accounted for 85% of all reads from this family (16 total ASVs). Of the 13 DA Vibrionaceae ASVs that were primarily restricted to the parrotfishes, each was found to be DA in only one parrotfish species (figure 4a). Likewise, the only Vibrionaceae ASV that was DA in surgeonfishes (ASV114) was dominant in A. tractus but not in A. coeruleus (figure 4a). The marked differences observed

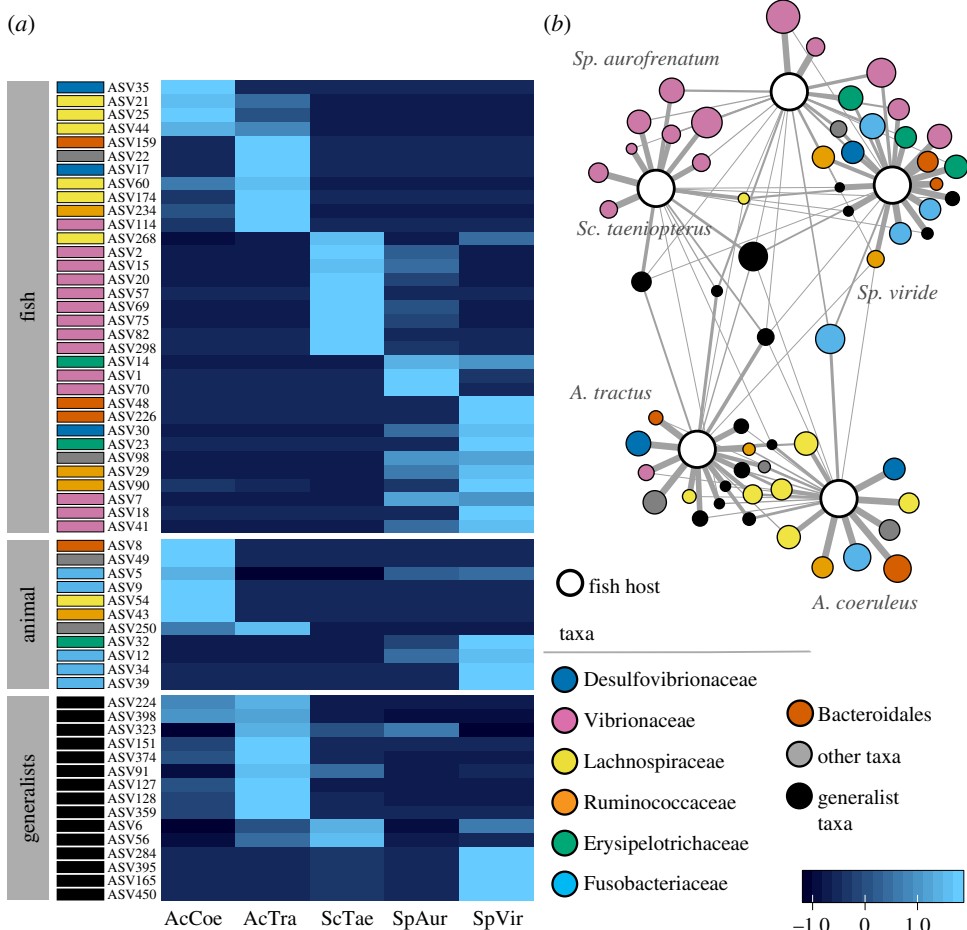

**Figure 4.** Distribution of each dominant microbiome member. (*a*) Heatmap of read abundance for each DA ASV (row) found within each of the five fish species (columns). DA ASVs are organized first by putative habitat/lifestyle category and second by the fish species in which they were DA. (*b*) Network diagram, where nodes are either DA ASVs or one of the five host fish species. DA ASV node size is proportional to log-transformed read abundance, and edge thickness is weighted by the abundance of that DA ASV found within a given fish. Edges were filtered to exclude weights less than 0.2. Colour for both panels represents microbial taxa. Putative generalist taxa are grouped together, coloured black, and represent all DA ASVs classified as Alphaproteobacteria, Rubritaleaceae, Flavobacteriaceae and Pirellulaceae. (Online version in colour.)

among herbivore species with respect to the putative symbionts they harbour—a pattern exemplified in a network analysis (figure 4*b*)—could be either a partial cause or a consequence of the niche differentiation that has transpired among these herbivores over the last 12 million years [35]. Regardless, such patterns indicate that these symbiont–host relationships are conserved along host evolutionary lines.

In contrast with putative resident symbionts, DA ASVs thought to be environmental generalists were not significantly associated with the feeding behaviour (Mantel $r = 0.596$, $p = 0.083$) or phylogeny (Mantel $r = 0.831$, $p = 0.075$) of the herbivore, nor were they associated with herbivore feeding behaviour after controlling for fish phylogeny (Mantel partial $r = 0.402$, $p = 0.150$). That putative environmental generalists were not strongly associated with either herbivore ecology or phylogeny perhaps should not be surprising, given that these DA ASVs were not Cyanobacteria (known prey of parrotfishes [56]) and could in many cases be surface-bound microbes that were incidentally ingested by the fish while feeding on other, macroscopic prey.

## 4. Discussion

Our study adds to a small but growing body of literature that indicates coral reef herbivorous fishes possess

species-specific intestinal microbiomes [38,57]. As a step forward, we establish that this species specificity is reflected in both the putative symbionts that these herbivores harbour and the environmental microbes that they incidentally or selectively ingest (figures 2–4). We go on to reveal that the identities of the putative microbial symbionts found in each herbivore vary as a function of the herbivore's evolutionary lineage. Thus, in addition to these herbivorous fishes having evolved unique morphologies [35] and feeding behaviours (figure 1), it appears that microbial symbioses within the intestine have also become functionally differentiated.

Growing evidence suggests that resident microbes within the intestine can aid in prey digestion and thus may underpin some of the feeding behaviours displayed by these herbivores. For example, the Lachnospiraceae ASVs, which we found in abundance only in the intestines of *A. coeruleus* and *A. tractus*, belong to a family of giant enteric microbes that are known to help Indo-Pacific surgeonfishes digest and assimilate brown algal polysaccharides [41]. Lachnospiraceae microbes likely benefit *A. coeruleus* and *A. tractus* in a similar way, given that these fishes also selectively consume brown macroalgae (figure 1) and must digest and assimilate the algae's carbohydrate-rich content [58]. Resident microbes within *Acanthurus* may also help detoxify the potent chemical defences produced by *Dictyota* and other brown seaweeds [59]; symbiotic bacteria within the intestines of mammalian [60] and insect [61]

**Table 1.** Distribution of DA ASVs across putative habitat preference.

| putative habitat | no. of ASVs[a] | perfect BLAST hits[b] | total IMNGS hits[c] | median IMNGS hits | taxa (total ASVs)[d] |
|---|---|---|---|---|---|
| fish | 33 | 2 | 430 | 8 | Bacteroidales (3), Brevinemataceae (1), Desulfovibrionaceae (3), Erysipelotrichaceae (2), Lachnospiraceae (6), Mollicutes (1), Ruminococcaceae (3), Vibrionaceae (14) |
| animal | 11 | 1 | 9934 | 180 | Bacteroidales (1), Erysipelotrichaceae (1), Fusobacteriaceae (5), Lachnospiraceae (1), Peptostreptococcaceae (1), Ruminococcaceae (1), Family XIII (1) |
| generalists | 15 | 10 | 9493 | 185 | Alphaproteobacteria (7), Flavobacteriaceae (2), Pirellulaceae (2), Rubritaleaceae (4) |

[a]Total number of DA ASVs from each putative habitat category.
[b]Number of ASVs with at least one perfect (100%) BLAST hit.
[c]Total number of IMNGS samples hit by ASVs.
[d]Taxonomic breakdown and total number of DA ASVs (in parentheses) for each putative habitat category.

herbivores help them to degrade secondary metabolites, enabling them to feed on toxic plants. As another possible example, the Erysipelotrichaceae, which we found in abundance only in *Sp. viride* and *Sp. aurofrenatum*, belong to a family of microbes that metabolize lipids in the human intestine [62]. Scraping parrotfishes appear to target microbial autotrophs within the reef matrix, which are much higher in lipid and protein content than their macroalgal counterparts [56]. Erysipelotrichaceae may thus help these parrotfishes to digest and assimilate their prey. Another group of ASVs that differentiated herbivores was the Vibrionaceae—a bacterial family that perform a variety of functions in nature [63,64] and are known to associate with fish intestines [65]. However, their putative functions in the intestine are unclear. Metagenomic analyses and studies that experimentally elucidate the functions of intestinal microbes [41] are particularly needed to disentangle which microbiome members are drivers versus passengers of herbivorous fish feeding behaviour.

Interestingly, while none of the putative residents we describe have been previously identified from Caribbean fishes, in virtually every case their closest relative in the database was a microbe isolated from the intestines of herbivorous coral reef fishes found in other parts of the world, namely the tropical Pacific and Red Sea (figure 3; electronic supplementary material, table S6). Thus, similar putative symbionts exist in the intestines of ecologically related herbivores that are separated by thousands of kilometres and millions of years of evolution [25,35]. Microbiome similarities could exist because of vertical transmission (from a common ancestor) or because closely related fishes perform similar ecological roles and thereby horizontally acquire symbionts from the environment in a similar way. Either mechanism appears feasible, as herbivore functional capabilities are generally conserved along phylogenetic lines and most herbivore prey (algae, coral) families are pantropical in distribution. More broadly, there also appears to have been convergent evolution of intestinal microbes in fishes and mammals [48], which speaks to why we also identified a small number of putative resident symbionts whose closest database hits were microbes isolated from the intestinal tracts of terrestrial mammals such as humans, monkeys, wild boar, gazelle and mice (figures 3 and 4). Exploring how herbivore microbiomes initially develop, and the degree to which they are shaped by local abiotic and biotic factors [48,66] is clearly a topic in need of further research across disparate ecosystems. Furthermore, herbivorous fishes often display ontogenetic shifts in diet [28,42,45] as well as significant flexibility in foraging behaviour among locations [32,45]. Targeted studies that link these ecological shifts to changes in their gut microbiomes should also be a priority for future studies.

It is often assumed that, because coral reefs are highly speciose, functional redundancies should be common within each trophic level. However, recent analyses of organismal traits have revealed that functional diversity within the coral reef fish assemblage is remarkably high and redundancies are surprisingly few [18], creating a situation in which functional diversity is particularly sensitive to human impacts such as fishing [19,20]. Our observations that Caribbean herbivores show little overlap in functional traits—in terms of the algae they feed on, where they feed from and the putative intestinal symbionts they harbour (figures 1–4)—supports and expands on this view. Our study indicates that the Caribbean herbivore assemblage possesses much higher levels of functional trait diversity than previously recognized and suggests that the loss of any given species will erode the functional potential of the community. Moreover, because the intestinal microbiome is proving to be a key determinant of how and why herbivores consume certain seaweeds [41], future studies should include the microbiome as a core organismal trait when assessing the functional potential of the assemblage.

Data accessibility. All data used or generated in this study are publicly available. Raw 16S rRNA sequences were deposited at the European Nucleotide Archive under the study accession number PRJEB28397 (ERP110594). See the electronic supplementary material, for complete details. Additional data, data products, reproducible workflows and code are available from our figshare project site (http://bit.ly/2MLJx1G) and on the project website (https://projectdigest.github.io/).

Authors' contributions. J.J.S. and D.B.R. conceived and initiated the study; J.J.S., D.E.B. and D.B.R. designed the study; T.C.A. and A.D. collected the field data and samples; J.J.S. and T.C.A. conducted the statistical analyses; J.J.S. and D.B.R. wrote the manuscript, with contributions and revisions from all authors.

Competing interests. The authors declare no competing interests.

Funding. This work was supported by the Bigelow Laboratory for Ocean Sciences and the Gordon and Betty Moore Foundation.

Acknowledgements. We thank the Florida Keys National Marine Sanctuary for permissions to conduct our research (grant no. FKNMS-2017-149), as well as Laura Palma and Elizabeth Lago for assistance in the field.

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
