## [Reviewer comments · Proceedings of the Royal Society B: Biological Sciences]

Review History

RSPB-2019-2367.R0 (Original submission)

Review form: Reviewer 1

Recommendation

Major revision is needed (please make suggestions in comments)

Scientific importance: Is the manuscript an original and important contribution to its field?

Excellent

General interest: Is the paper of sufficient general interest?

Excellent

Quality of the paper: Is the overall quality of the paper suitable?

Good

Is the length of the paper justified?

Yes

Should the paper be seen by a specialist statistical reviewer?

Yes

Do you have any concerns about statistical analyses in this paper? If so, please specify them explicitly in your report.

Yes

It is a condition of publication that authors make their supporting data, code and materials available - either as supplementary material or hosted in an external repository. Please rate, if applicable, the supporting data on the following criteria.

Is it accessible?

Yes

Is it clear?

Yes

Is it adequate?

Yes

Do you have any ethical concerns with this paper?

No

Comments to the Author

This is a nice paper that provides new and interesting data regarding fish microbiomes, with valuable insights into gut microbiome specialization related to fish diet, adding to the still slim microbiome research on wild fish on this subject. I have various comments that could help improve the manuscript.

Introduction:

- I think paragraphs 2 and 3 need to be clearer regarding what is known from herbivorous species and what was specifically determined for fish herbivours.

Methods:

- there were observations of fish foraging behaviour in two separate years, but was there any inter-annual variation in these behaviours? Figure 1 depicts marked differences between most species but it does not show whether there were differences between years.

- I find the description of what is published for the diet of these fish too simplistic given the existing literature on the subject. For example, similarly to what is presented here, Duran et al (2019) show changes on feeding behaviour of the two *Acanthurus* species between the studied reefs (I am assuming the data is not the same as the one presented here) as well as some ontogenetic shifts, including differences in animal food uptake. Similarly, for *S. viride* other studies have suggested ontogenetic changes to feeding behavior (Rooij et al. 1995 MEPS; Bruggemann et al. 1994. MEPS). The work of Catano et al (2016) is also important for the context of this paper since they suggest that the feeding behavior of herbivours can differ amongst the studied localities. It would be important to describe this plasticity and whether it would affect the conclusions. It's also important that the authors explain whether the habitat at Pickles Reef is different from the sites where foraging behavior was studied, and explicitly state whether or not this could affect their conclusions. Also, since ontogenetic differences were found/suggested to occur for some of the studied species, did the authors conduct any analysis on the microbiome/behavior to test for this? Also, why is the used terminology to define life-stage different between fish species?

- More complex (e.g. GLM) models could be use to test the differences between species in alpha diversity indices which are not normally distributed

- One of the conclusions of the paper is that fish harbor microbiomes with differing functionality, but state that despite using a popular bioinformatic package, they were unable to infer specific functions to these fish microbiomes. Which package was this? Was it Picrust? Which version? Picrust2 may work better (see

<https://www.biorxiv.org/content/biorxiv/early/2019/06/15/672295.full.pdf>)

- Finally, as far as I understood, microbiome data was generated in different Illumina runs. Did

the authors use any mock community to standardize/compare biases between sequencing runs? Did they conduct any other analysis to assess potential differences?

Review form: Reviewer 2

Recommendation

Accept with minor revision (please list in comments)

Scientific importance: Is the manuscript an original and important contribution to its field?

Good

General interest: Is the paper of sufficient general interest?

Good

Quality of the paper: Is the overall quality of the paper suitable?

Good

Is the length of the paper justified?

Yes

Should the paper be seen by a specialist statistical reviewer?

No

Do you have any concerns about statistical analyses in this paper? If so, please specify them explicitly in your report.

No

It is a condition of publication that authors make their supporting data, code and materials available - either as supplementary material or hosted in an external repository. Please rate, if applicable, the supporting data on the following criteria.

Is it accessible?

Yes

Is it clear?

Yes

Is it adequate?

Yes

Do you have any ethical concerns with this paper?

No

Comments to the Author

The manuscript entitled "Intestinal microbes: an axis of functional diversity among large marine consumers" describes that intestinal microbiome is potentially important as an indicator of functional trait herbivorous fishes, vital to the ecology of coral reefs.

It is an interesting paper, a study necessary for the specific field. The document contains new and interesting results and appears to be the first major effort in this type of subject. The experiments are very straight forward; both the introduction and methods were clearly written, and I appreciated the Figures. The abundant taxa may play an important role in shaping the bacterial communities as they were surveyed.

There are some points that should be considered by the authors before publication of the manuscript.

1. Community structure. As it is certainly possible that niche may select for certain bacterial groups, this is the first report of their dominance after enrichment. This is pointed out by the authors, but they should consider one of their treatments that may have exacerbated this effect. Unlike other molecular surveys of bacterial diversity, the authors did not survey the original sample (water column, bed sediment etc). Unfortunately, core samples were not surveyed using the sequencing-based approach. It would have been an interesting control for their sample handling mentioned above. The authors need to consider addressing this possibility in their discussion.

The introduction section should be more structured on important issues that have been answered (or not) for this type of environment and bring the reader to the overall goal.

Discussion: Although I agree with most of what was inferred by the authors, I think caution should be taken when discussing functionality (even potential) with respect to data obtained with functional markers (16S). I suggest that the authors highlight the problems and pitfalls that may occur in such analyses.

Minor comments:

Lines 35-37: The last point indicated in the abstract was not clearly shown in this paper. Reformulate the last sentence of the Abstract, because with your analysis you found that the microbiome was related with the phylogeny and not with the feeding behaviors, and also, you did not analyze the microbiome functionalities to say that may confer important information regarding the digestive capabilities. Add some real conclusion of your work.

Lines 68-70: you add information regarding the transients vs residents microbiome, but you did not perform this type of analysis and/or achieved this results. I suggest removing this sentence and add information that it is directly relevant to your study.

Lines 226-229: Reformulate the sentence. To make your first results' sentence stronger, invert the information... firstly add the results and then the analysis' tool used.

Line 328: change "with respect" to "regarding" or another word.

Lines 412-417: Modify this part. I suggest to improve the conclusion, answering the objectives of your work.

Lines 670-675: Improvement of the Figure 2. I suggest adding spaces between the figures a, b, c, to improve the visualization. Also, indicate the statistical differences on the Shannon index; it is on the next but not in the figure.

Review the reference citation through the text.

Decision letter (RSPB-2019-2367.R0)

31-Dec-2019

Dear Dr Scott:

Your manuscript has now been peer reviewed and the reviews have been assessed by an

Associate Editor. The reviewers' comments (not including confidential comments to the Editor) and the comments from the Associate Editor are included at the end of this email for your reference. As you will see, the reviewers and the Editors have raised some concerns with your manuscript and we would like to invite you to revise your manuscript to address them.

Research ethics:

Use of animals and field studies:

Please submit a copy of your revised paper within three weeks. If we do not hear from you within this time your manuscript will be rejected. If you are unable to meet this deadline please let us know as soon as possible, as we may be able to grant a short extension.

Best wishes,
Dr Daniel Costa
mailto:proceedingsb@royalsociety.org

Associate Editor
Board Member: 1
Comments to Author:

The reviewers and handling editor concur that the manuscript provides timely, highly relevant insights, and appears to be among the first to document the relationships between intestinal flora and megafaunal grazing etc. The clarity of the figures and the writing were very much appreciated, and ensured that the points raised by the authors were (for the most part) clear and concise.

There are a number of issues that remain to be addressed prior to the manuscript being accepted. The points below are some of the most salient concerns raised by the two reviewers, and thus warrant the most attention:

1. Clarification about the community structure and the potential for confounding factors: As it is certainly possible that niche may select for certain bacterial groups, this is the first report of their dominance after enrichment. This is pointed out by the authors, but they should consider one of their treatments that may have exacerbated this effect. Unlike other molecular surveys of bacterial diversity, the authors did not survey the original sample (water column, bed sediment etc). Unfortunately, core samples were not surveyed using the sequencing-based approach. It would have been an interesting control for their sample handling mentioned above. The authors need to consider addressing this possibility in their discussion.

Paragraphs 2 and 3 need to be clearer regarding what is known from herbivorous species and what was specifically determined for fish herbivores.

The representation of the existing literature in this manuscript is modest at best, and some of the representations about the diet of these fishes may be oversimplified. As the reviewer notes, "For example, similarly to what is presented here, Duran et al (2019) show changes on feeding behaviour of the two *Acanthurus* species between the studied reefs (I am assuming the data is not the same as the one presented here) as well as some ontogenetic shifts, including differences in animal food uptake. Similarly, for *S. viride* other studies have suggested ontogenetic changes to feeding behavior (Rooij et al. 1995 MEPS; Bruggemann et al. 1994. MEPS). The work of Catano et al (2016) is also important for the context of this paper since they suggest that the feeding behavior of herbivours can differ amongst the studied localities. It would be important to describe this plasticity and whether it would affect the conclusions."

"It's also important that the authors explain whether the habitat at Pickles Reef is different from the sites where foraging behavior was studied, and explicitly state whether or not this could affect their conclusions."

Please revise accordingly to address these aforementioned issues, as well as those outlined in the reviews.

Reviewer(s)' Comments to Author:

Referee: 1

Comments to the Author(s)

This is a nice paper that provides new and interesting data regarding fish microbiomes, with valuable insights into gut microbiome specialization related to fish diet, adding to the still slim microbiome research on wild fish on this subject. I have various comments that could help improve the manuscript.

Introduction:

- I think paragraphs 2 and 3 need to be clearer regarding what is known from herbivorous species and what was specifically determined for fish herbivours.

Methods:

- there were observations of fish foraging behaviour in two separate years, but was there any inter-annual variation in these behaviours? Figure 1 depicts marked differences between most species but it does not show whether there were differences between years.

- I find the description of what is published for the diet of these fish too simplistic given the existing literature on the subject. For example, similarly to what is presented here, Duran et al (2019) show changes on feeding behaviour of the two *Acanthurus* species between the studied reefs (I am assuming the data is not the same as the one presented here) as well as some ontogenetic shifts, including differences in animal food uptake. Similarly, for *S. viride* other studies have suggested ontogenetic changes to feeding behavior (Rooij et al. 1995 MEPS; Bruggemann et al. 1994. MEPS). The work of Catano et al (2016) is also important for the context of this paper since they suggest that the feeding behavior of herbivours can differ amongst the studied localities. It would be important to describe this plasticity and whether it would affect the conclusions. It's also important that the authors explain whether the habitat at Pickles Reef is different from the sites where foraging behavior was studied, and explicitly state whether or not this could affect their conclusions. Also, since ontogenetic differences were found/suggested to occur for some of the studied species, did the authors conduct any analysis on the microbiome/behavior to test for this? Also, why is the used terminology to define life-stage different between fish species?

- More complex (e.g. GLM) models could be used to test the differences between species in alpha diversity indices which are not normally distributed

- One of the conclusions of the paper is that fish harbor microbiomes with differing functionality, but state that despite using a popular bioinformatic package, they were unable to infer specific functions to these fish microbiomes. Which package was this? Was it Picrust? Which version?

Picrust2 may work better (see

<https://www.biorxiv.org/content/biorxiv/early/2019/06/15/672295.full.pdf>)

- Finally, as far as I understood, microbiome data was generated in different Illumina runs. Did the authors use any mock community to standardize/compare biases between sequencing runs? Did they conduct any other analysis to assess potential differences?

Referee: 2

Comments to the Author(s)

The manuscript entitled "Intestinal microbes: an axis of functional diversity among large marine consumers" describes that intestinal microbiome is potentially important as an indicator of functional trait herbivorous fishes, vital to the ecology of coral reefs.

It is an interesting paper, a study necessary for the specific field. The document contains new and interesting results and appears to be the first major effort in this type of subject. The experiments are very straight forward; both the introduction and methods were clearly written, and I appreciated the Figures. The abundant taxa may play an important role in shaping the bacterial communities as they were surveyed.

There are some points that should be considered by the authors before publication of the manuscript.

1. Community structure. As it is certainly possible that niche may select for certain bacterial groups, this is the first report of their dominance after enrichment. This is pointed out by the authors, but they should consider one of their treatments that may have exacerbated this effect. Unlike other molecular surveys of bacterial diversity, the authors did not survey the original sample (water column, bed sediment etc). Unfortunately, core samples were not surveyed using the sequencing-based approach. It would have been an interesting control for their sample handling mentioned above. The authors need to consider addressing this possibility in their discussion.

The introduction section should be more structured on important issues that have been answered (or not) for this type of environment and bring the reader to the overall goal.

Discussion: Although I agree with most of what was inferred by the authors, I think caution should be taken when discussing functionality (even potential) with respect to data obtained with functional markers (16S). I suggest that the authors highlight the problems and pitfalls that may occur in such analyses.

Minor comments:

Lines 35-37: The last point indicated in the abstract was not clearly shown in this paper. Reformulate the last sentence of the Abstract, because with your analysis you found that the microbiome was related with the phylogeny and not with the feeding behaviors, and also, you did not analyze the microbiome functionalities to say that may confer important information regarding the digestive capabilities. Add some real conclusion of your work.

Lines 68-70: you add information regarding the transients vs residents microbiome, but you did not perform this type of analysis and/or achieved this results. I suggest removing this sentence and add information that it is directly relevant to your study.

Lines 226-229: Reformulate the sentence. To make your first results' sentence stronger, invert the information... firstly add the results and then the analysis' tool used.

Line 328: change "with respect" to "regarding" or another word.

Lines 412-417: Modify this part. I suggest to improve the conclusion, answering the objectives of your work.

Lines 670-675: Improvement of the Figure 2. I suggest adding spaces between the figures a, b, c, to improve the visualization. Also, indicate the statistical differences on the Shannon index; it is on the next but not in the figure.

Review the reference citation through the text.

Author's Response to Decision Letter for (RSPB-2019-2367.R0)

See Appendix A.

RSPB-2019-2367.R1 (Revision)

Review form: Reviewer 1

Recommendation

Major revision is needed (please make suggestions in comments)

Scientific importance: Is the manuscript an original and important contribution to its field?

Excellent

General interest: Is the paper of sufficient general interest?

Excellent

Quality of the paper: Is the overall quality of the paper suitable?

Good

Is the length of the paper justified?

Yes

Should the paper be seen by a specialist statistical reviewer?

No

Do you have any concerns about statistical analyses in this paper? If so, please specify them explicitly in your report.

No

It is a condition of publication that authors make their supporting data, code and materials available - either as supplementary material or hosted in an external repository. Please rate, if applicable, the supporting data on the following criteria.

Is it accessible?

Yes

Is it clear?

Yes

Is it adequate?

Yes

Do you have any ethical concerns with this paper?

No

Comments to the Author

The authors made overall satisfactory changes to the manuscript, however I must maintain that the known ontogenetic diet shifts for these fish species should be contextualized in the manuscript. I understand the authors' point of view that they are looking at the general diet patterns commonly reported for these species, but the fact that there are known variations in diet preferences related to age (and perhaps even locality, Catano et al 2016) is also part of the state of the art. This knowledge is based on substantial research (to which the authors' themselves contributed), and importantly, their sampling DOES include fish from different life stages. Therefore, I also don't think omitting the original data on fish life-stage is correct. If the data were imprecise for surgeonfish please correct them (e.g. with fish size).

Research in this field is growing fast and in the future other researchers might want to use these data to tackle potentially more subtle effects of these reported dietary shifts. This could be easily achieved by using a GLMM and a random age/life-stage effect. Therefore, if the authors do not wish to test this age effect in the present manuscript, I strongly suggest the original metadata is kept and that the authors provide that background in Discussion.

Decision letter (RSPB-2019-2367.R1)

11-Feb-2020

Dear Dr Scott:

Your manuscript has now been peer reviewed and the reviews have been assessed by an Associate Editor. The reviewers' comments (not including confidential comments to the Editor) and the comments from the Associate Editor are included at the end of this email for your reference. As you will see, the reviewers and the Editors have raised some concerns with your manuscript and we would like to invite you to revise your manuscript to address them.

Normally I would reject a manuscript after a round of review. However, the reviewer is positive about your manuscript, but provides important recommendation that would improve the manuscript. As you have already gone through one round of reject and resubmit, I am therefore providing you one final opportunity to revise the manuscript accordingly. If deemed necessary by the Associate Editor, your manuscript will be sent back to one or more of the original reviewers for assessment. If the original reviewers are not available we may invite new reviewers. Please note that we cannot guarantee eventual acceptance of your manuscript at this stage.

Research ethics:

Use of animals and field studies:

Please submit a copy of your revised paper within three weeks. If we do not hear from you

within this time your manuscript will be rejected. If you are unable to meet this deadline please let us know as soon as possible, as we may be able to grant a short extension.

Best wishes,
Dr Daniel Costa
Editor, Proceedings B
mailto:proceedingsb@royalsociety.org

Reviewer(s)' Comments to Author:

Referee: 1

Comments to the Author(s)

The authors made overall satisfactory changes to the manuscript, however I must maintain that the known ontogenetic diet shifts for these fish species should be contextualized in the manuscript. I understand the authors' point of view that they are looking at the general diet patterns commonly reported for these species, but the fact that there are known variations in diet preferences related to age (and perhaps even locality, Catano et al 2016) is also part of the state of the art. This knowledge is based on substantial research (to which the authors' themselves contributed), and importantly, their sampling DOES include fish from different life stages. Therefore, I also don't think omitting the original data on fish life-stage is correct. If the data were imprecise for surgeonfish please correct them (e.g. with fish size).

Research in this field is growing fast and in the future other researchers might want to use these data to tackle potentially more subtle effects of these reported dietary shifts. This could be easily achieved by using a GLMM and a random age/life-stage effect. Therefore, if the authors do not wish to test this age effect in the present manuscript, I strongly suggest the original metadata is kept and that the authors provide that background in Discussion.

Author's Response to Decision Letter for (RSPB-2019-2367.R1)

See Appendix B.

RSPB-2019-2367.R2 (Revision)

Review form: Reviewer 1

Recommendation

Accept as is

Scientific importance: Is the manuscript an original and important contribution to its field?

Excellent

General interest: Is the paper of sufficient general interest?

Excellent

Quality of the paper: Is the overall quality of the paper suitable?

Excellent

Is the length of the paper justified?

Yes

Should the paper be seen by a specialist statistical reviewer?

No

Do you have any concerns about statistical analyses in this paper? If so, please specify them explicitly in your report.

No

It is a condition of publication that authors make their supporting data, code and materials available - either as supplementary material or hosted in an external repository. Please rate, if applicable, the supporting data on the following criteria.

Is it accessible?

Yes

Is it clear?

Yes

Is it adequate?

Yes

Do you have any ethical concerns with this paper?

No

Comments to the Author

I am now satisfied with this final version and I congratulate the authors for the good work.

Decision letter (RSPB-2019-2367.R2)

10-Mar-2020

Dear Dr Scott

I am pleased to inform you that your manuscript entitled "Intestinal microbes: an axis of functional diversity among large marine consumers" has been accepted for publication in Proceedings B.

Open Access

Paper charges

Sincerely,

Dr Daniel Costa

Appendix A

Jarrold J. Scott
Smithsonian Tropical Research Institute
Apartado 0843-03092
Balboa, República de Panamá
Phone: +507 6733-6268
E-mail: jarrod.jude.scott@gmail.com

January 20, 2020

Proceedings of the Royal Society: Biological Sciences

Dear Drs. Costa & Singh-Shepherd,

Please find enclosed a revision of our manuscript entitled, *Intestinal microbes: an axis of functional diversity among large marine consumers* for your continued consideration as a Research Article in *Proceedings of the Royal Society: Biological Sciences*.

On behalf of my co-authors, I would like to thank you for the opportunity to revise our manuscript. The comments and suggestions provided by the reviewers were well-reasoned. By responding to the reviewer concerns, our manuscript is now notably improved. We also appreciate the summary comments provided by the Associate Editor. The summary helped us to identify the most salient concerns raised by the reviewers.

Attached below you will find a detailed, point-by-point response (in red) to the reviewer comments. Please note that we did not respond directly to the Editor's summary, since by responding to each reviewer comment, we simultaneously address the Editor's summary. We hope this is acceptable.

Again, thank you for your continued interest in our manuscript.

Sincerely on behalf of all
co-authors,

Jarrold J. Scott, PhD
Bocas del Toro, Panamá

Editor Summary: The reviewers and handling editor concur that the manuscript provides timely, highly relevant insights, and appears to be among the first to document the relationships between intestinal flora and megafaunal grazing etc. The clarity of the figures and writing were very much appreciated, and ensured that the points raised by the authors were (for the most part) clear and concise.

There are a number of issues that remain to be addressed prior to the manuscript being accepted. The points below are some of the most salient concerns raised by the two reviewers, and thus warrant the most attention:

1. Clarification about the community structure and the potential for confounding factors: As it is certainly possible that niche may select for certain bacterial groups, this is the first report of their dominance after enrichment. This is pointed out by the authors, but they should consider one of their treatments that may have exacerbated this effect. Unlike other molecular surveys of bacterial diversity, the authors did not survey the original sample (water column, bed sediment etc). Unfortunately, core samples were not surveyed using the sequencing-based approach. It would have been an interesting control for their sample handling mentioned above. The authors need to consider addressing this possibility in their discussion.

Paragraphs 2 and 3 need to be clearer regarding what is known from herbivorous species and what was specifically determined for fish herbivores.

The representation of the existing literature in this manuscript is modest at best, and some of the representations about the diet of these fishes may be oversimplified. As the reviewer notes, "For example, similarly to what is presented here, Duran et al (2019) show changes on feeding behaviour of the two *Acanthurus* species between the studied reefs (I am assuming the data is not the same as the one presented here) as well as some ontogenetic shifts, including differences in animal food uptake. Similarly, for *S. viride* other studies have suggested ontogenetic changes to feeding behavior (Rooij et al. 1995 MEPS; Bruggemann et al. 1994. MEPS). The work of Catano et al (2016) is also important for the context of this paper since they suggest that the feeding behavior of herbivours can differ amongst the studied localities. It would be important to describe this plasticity and whether it would affect the conclusions."

"It's also important that the authors explain whether the habitat at Pickles Reef is different from the sites where foraging behavior was studied, and explicitly state whether or not this could affect their conclusions."

Please revise accordingly to address these aforementioned issues, as well as those outlined in the reviews.

To capture both the summary provided by the Editor and the Reviewer's comments, we respond below to each point-by-point comments raised by each reviewer.

Referee: 1 This is a nice paper that provides new and interesting data regarding fish microbiomes, with valuable insights into gut microbiome specialization related to fish diet, adding to the still slim microbiome research on wild fish on this subject. I have various comments that could help improve the manuscript.

We thank R1 for their positive and constructive feedback. We hope they agree that our manuscript is now improved, and that we have addressed their concerns/suggestions in a satisfactory manner.

Introduction: I think paragraphs 2 and 3 need to be clearer regarding what is known from herbivorous species and what was specifically determined for fish herbivours.

We revised paragraphs 2 and 3 to clarify that we are specifically referring to coral reef fishes rather than general coral reef herbivores (i.e., herbivorous fishes and sea urchins). We appreciate that our original text was unclear, and we hope that this modification makes our Introduction more precise.

Methods: There were observations of fish foraging behaviour in two separate years, but was there any inter-annual variation in these behaviours? Figure 1 depicts marked differences between most species but it does not show whether there were differences between years.

Unfortunately our study design did not allow us to address this question, as the parrotfishes were observed in 2014 and surgeonfishes in 2016. We apologize for not making this clear in our original submission. We have added text to clarify this point (line 110). Given that observations were made at the same sites and under similar conditions (e.g., during the same season when algal assemblages and temperatures were similar) we do not believe that any of the differences we observed were driven by inter-annual variation. In addition, our previous work in this ecosystem indicates that while each species does exhibit plasticity in foraging behavior, this plasticity is small compared to the large interspecific differences we have observed (for example, see Figs. S6, S7, S9, and S10 in Adam et al. 2018 and Figs. 3, 4, and 6 in Duran et al. 2019).

I find the description of what is published for the diet of these fish too simplistic given the existing literature on the subject. For example, similarly to what is presented here, Duran et al (2019) show changes on feeding behaviour of the two *Acanthurus* species between the studied reefs (I am assuming the data is not the same as the one presented here) as well as some ontogenetic shifts, including differences in animal food uptake. Similarly, for *S. viride* other studies have suggested ontogenetic changes to feeding behavior (Rooij et al. 1995 MEPS; Bruggemann et al. 1994. MEPS). The work of Catano et al (2016) is also important for the context of this paper since they suggest that the feeding behavior of herbivours can differ amongst the studied localities. It would be important to describe this plasticity and whether it would affect the conclusions.

We agree that each of these species exhibit ontogenetic shifts in foraging behavior and that these shifts are important for understanding niche partitioning among species. In fact, we have described many of these shifts in great detail in recently published work (see Figs. 5, S8, S11, and S12 in Adam et al. 2018 and Fig. 5 in Duran et al. 2019). Unfortunately our sample size for the microbial analyses were simply not large enough to look at ontogenetic shifts in gut microbiomes. Thus in this manuscript, we were unable to incorporate the potential impacts of ontogeny on gut microbiomes or foraging ecology. Given our inability to address ontogenetic shifts or inter-site variability in gut microbiomes in this paper, we instead focused on briefly summarizing what is known about the foraging ecology of these species. We tried to do this in a way that was general enough to provide important context without going into detail about ontogenetic shifts in diet or plasticity in foraging ecology. We would like to emphasize that these generalizations about each species are accurate across a wide range of size classes and reef types. By studying the foraging behavior of these fishes at multiple sites, we achieved a general description of their foraging behavior that we can use to test whether differences in microbiomes are related to between-species differences in foraging behavior.

It's also important that the authors explain whether the habitat at Pickles Reef is different from the sites where foraging behavior was studied, and explicitly state whether or not this could affect their conclusions. Also, since ontogenetic differences were found/suggested to occur for some of the studied species, did the authors conduct any analysis on the microbiome/behavior to test for this?

This is a good point. We added a brief description of the habitat at Pickles Reef (lines 144–150) as well as the reefs where the behavioral work was conducted (Molasses, French, and Conch Reefs) (lines 110–112). Briefly, all reefs have similar benthic assemblages (low coral cover and high cover of turf and macroalgae—especially *Dictyota*). However, the reefs do vary in architectural complexity. Molasses and French are high-relief reefs, Conch is a low-relief reef and Pickles is intermediate. Because previous work (Adam et al. 2018) has shown that these herbivore species partition resources similarly among sites, the fact that the behavioral observations were conducted at different sites than the microbiome work is unlikely to affect our ability to link differences in fish microbiomes with differences in their foraging behavior. As stated above, we did not have enough samples to test for potential effects of ontogeny on gut microbiomes. While it would be interesting to see how the microbiomes change with ontogeny, our observation that each species has a highly distinct microbiome suggests that any ontogenetic effects are small compared to the differences we observed among species (at least for the size ranges of fishes we studied).

Also, why is the used terminology to define life-stage different between fish species?

In our original submission, we included a "life phase" column in the table to describe differences in life stages of each individual used in the study (Table S3). Parrotfishes are sequential hermaphrodites with distinct sexually dimorphic color phases. The terms 'initial phase' and 'terminal phase' refer to these two distinct phases. Surgeonfishes, in contrast, are not sequential hermaphrodites. The surgeonfish categories were selected based on the size of each animal. Thanks in part to the reviewer's comment, we now believe that our categories for the surgeonfish were arbitrary and imprecise. We have therefore elected to eliminate the categorical designations for the surgeonfish. **Table S3 has been updated accordingly.**

More complex (e.g. GLM) models could be use to test the differences between species in alpha diversity indices which are not normally distributed

Apologies if our approach was confusing. As noted in the original manuscript (lines 160–168), the Shannon diversity data were indeed normally distributed. Therefore, we analyzed differences in alpha (Shannon) diversity among species using ANOVA. Importantly, we also assessed the other diversity indices (which were not normally distributed) using a nonparametric equivalent to the ANOVA (a Kruskal-Wallis test). We added a sentence clarifying this on line 171–171. Analysis of the non-normally distributed indices revealed the exact same patterns (see associated supplementary materials under the subheading **Additional alpha diversity tests**). Given the relatively straightforward nature of these diversity data, we did not feel the need to employ more complex models such as GLMs. You can find the associated code and results at: https://projectdigest.github.io/4_diversity.html#alpha-diversity

One of the conclusions of the paper is that fish harbor microbiomes with differing functionality, but state that despite using a popular bioinformatic package, they were unable to infer specific functions to these fish microbiomes. Which package was this? Was it Picrust? Which version? Picrust2 may work better (see <https://www.biorxiv.org/content/biorxiv/early/2019/06/15/672295.full.pdf>)

Omitting these details was an oversight on our part and we appreciate the reviewer catching this. Yes, in the original manuscript we used PICRUS_t for the assessment and, per Reviewer 1's suggestion, we also ran PICRUS_t2 on these same data. In both case nearest-sequenced taxon index (NSTI) for all samples were above (in many cases well above) the recommended limits advised by the developers of PICRUS_t. The reviewer was correct—PICRUS_t2 did perform slightly

better, however NSTI values were still too high. Therefore we concluded that these tools were inappropriate for this dataset.

We decided to remove mention of inferring function using 16S data from the manuscript. First, we do not believe that 16S rRNA data is suitable for assessing the functional potential of microbial communities from most environments. True, the approach is more convenient and less expensive than other methods (e.g., metagenomics). However, as rightly pointed out by Reviewer 2, there are many pitfalls and problems that can occur with such analyses. For example Chevrette and colleagues analyzed *Streptomyces* genomes and found high variability of average nucleotide identity (ANI) and ortholog presence/absence in strains with identical 16S sequences (<https://www.frontiersin.org/articles/10.3389/fmicb.2019.02170/full>). Second, inferring community function from 16S data was not a goal of our study. Given that both reviewers expressed concern over inferring function from 16S data (a concern we agree with), we felt that inclusion of these analyses was confusing and detracted from our actual objectives.

Finally, as far as I understood, microbiome data was generated in different Illumina runs. Did the authors use any mock community to standardize/compare biases between sequencing runs? Did they conduct any other analysis to assess potential differences?

Another good point. Though we acknowledge the importance of mock communities we did not use one in our study. Where biases are concerned, we expect differences across runs. However, a particularly powerful part of the DADA2 workflow is that it is designed to handle projects spanning multiple runs. This is because assessing read quality, filtering, learning error rate, inferring sequence variants, and chimera checking all happen on a run-by-run basis. Error rate for example is learned from the run itself and thus the workflow is “tailored” to the peculiarities of each run. And since the output are exact sequence variants, datasets are directly comparable across runs without needing to pool the data. Only at the end of our DADA2 workflow were runs combined into a single dataset. (https://projectdigest.github.io/2_dada2.html).

Referee: 2

The manuscript entitled “Intestinal microbes: an axis of functional diversity among large marine consumers” describes that intestinal microbiome is potentially important as an indicator of functional trait herbivorous fishes, vital to the ecology of coral reefs.

It is an interesting paper, a study necessary for the specific field. The document contains new and interesting results and appears to be the first major effort in this type of subject. The experiments are very straight forward; both the introduction and methods were clearly written, and I appreciated the Figures. The abundant taxa may play an important role in shaping the bacterial communities as they were surveyed.

There are some points that should be considered by the authors before publication of the manuscript.

1. Community structure. As it is certainly possible that niche may select for certain bacterial groups, this is the first report of their dominance after enrichment. This is pointed out by the authors, but they should consider one of their treatments that may have exacerbated this effect. Unlike other molecular surveys of bacterial diversity, the authors did not survey the original sample (water column, bed sediment etc). Unfortunately, core samples were not surveyed using the sequencing-based approach. It would have been an interesting control for their sample handling mentioned above. The authors need to consider addressing this possibility in their discussion.

Apologies, but we are not certain what “this” is referencing, what “treatment” R2 is alluding to, or how any such treatment “may have exacerbated this effect”. We do our best to respond below.

In our manuscript, we reveal—through a LeFSE analysis—which intestinal microbes are differentially abundant within each fish’s microbiome. As explained previously in the Discussion (lines 393-403 of the original manuscript) it is impossible to determine how a fish acquires a differentially abundant (DA) microbe using this tool. DA microbes, including those hypothesized to be resident symbionts, could be vertically transmitted (i.e., passed from fish parent to offspring) or horizontally acquired (i.e., from the surrounding environment). Consequently, while we describe patterns of microbial community structure and hypothesize which microbes might be resident symbionts, we do not infer modes of acquisition, as multiple sources and acquisition mechanisms are possible.

Indeed, we did not simultaneously collect water and sediment samples. However, we do note that our bioinformatic analyses (see **Figures 3 and 4**) demark which DA microbes are highly similar to those commonly found in the environment vs. those that are only known from the intestines of herbivorous fishes. While not a true “control”, this information does shed some light on whether the DA taxa we identified are, or are not, likely to broadly occur in the ambient environment.

The introduction section should be more structured on important issues that have been answered (or not) for this type of environment and bring the reader to the overall goal.

We believe our Introduction, as currently written, clearly articulates the most important elements regarding what is known about our study system, what is unknown, and how our objectives help to close the knowledge gap (and pave the way for future research in this area). Given that the other reviewer and editor felt that our Introduction was clear and strong, we have not substantially altered this section. However, as noted above, we did make clarifications where necessary.

Discussion: Although I agree with most of what was inferred by the authors, I think caution should be taken when discussing functionality (even potential) with respect to data obtained with functional markers (16S). I suggest that the authors highlight the problems and pitfalls that may occur in such analyses.

Excellent point. To be clear, our Discussion of possible functions (lines 360-385 of the original submission), was not inferred using 16S data, but were rather derived from targeted functional studies in the literature. That said, it is important to note that functional inferences using 16S data (e.g., PICRUSt) have problems and pitfalls. As outlined above in our response to Reviewer 1’s comments, we decided to remove mention of inferring function using 16S data from the manuscript. First, we do not believe that 16S rRNA data is suitable for assessing the functional potential of microbial communities from most environments. True, the approach is more convenient and less expensive than other methods (e.g., metagenomics). However, as rightly pointed out by Reviewer 2, there are many pitfalls and problems that can occur with such analyses. For example Chevrette and colleagues analyzed *Streptomyces* genomes and found high variability of average nucleotide identity (ANI) and ortholog presence/absence in strains with identical 16S sequences (<https://www.frontiersin.org/articles/10.3389/fmicb.2019.02170/full>). Second, inferring community function from 16S data was not a goal of our study. Given that both reviewers expressed concern over inferring function from 16S data (a concern we agree with), we felt that inclusion of these analyses was confusing and detracted from our actual objectives.

Minor comments:

Lines 35-37: The last point indicated in the abstract was not clearly shown in this paper. Reformulate the last sentence of the Abstract, because with your analysis you found that the microbiome was related with the phylogeny and not with the feeding behaviors, and also, you did not analyze the microbiome functionalities to say that may confer important information regarding the digestive capabilities. Add some real conclusion of your work.

The last sentence has been modified to argue our main take-home point, which is that the intestinal microbiome should be considered as a potentially important functional trait among these large-bodied consumers.

Lines 68-70: you add information regarding the transients vs residents microbiome, but you did not perform this type of analysis and/or achieved this results. I suggest removing this sentence and add information that it is directly relevant to your study.

To be clear, this sentence is referencing the large gap in knowledge in this field, not what we specifically tested in our study. We describe the objectives of our study in the following paragraph.

Lines 226-229: Reformulate the sentence. To make your first results' sentence stronger, invert the information. . . firstly add the results and then the analysis' tool used.

Change made as advised. See lines 245-248

Line 328: change "with respect" to "regarding" or another word.

Change made as advised. See line 353

Lines 412-417: Modify this part. I suggest to improve the conclusion, answering the objectives of your work.

We felt that it was appropriate to end the manuscript by putting our findings into the context of larger questions in the field. Our goal here was to suggest how future studies should consider the microbiome as an important component of reef fish ecology—on par with other functional traits. Since that was our objective with these concluding remarks, we elected—with respect to the reviewer—to keep the text "as is".

Lines 670-675: Improvement of the Figure 2. I suggest adding spaces between the figures a, b, c, to improve the visualization. Also, indicate the statistical differences on the Shannon index; it is on the next but not in the figure.

Thank you for the suggestion. We modified the figure to give everything more breathing room and added letters in the top bar of panel "b" to indicate significant differences among host species. We modified the figure legend to reflect the change. We hope these changes make the figure more visually appealing and informative.

Review the reference citation through the text.

Reviewed as advised.

Appendix B

Jarrod J. Scott, PhD
Smithsonian Tropical Research Institute
Apartado 0843-03092
Balboa, Republica de Panama
E-mail: jarrod.jude.scott@gmail.com
Phone: +507 6733-6268

2020-03-03

Proceedings of the Royal Society: Biological Sciences

Dear Drs. Costa and Singh-Shepherd,

Please find enclosed a second revision of our manuscript entitled, "*Intestinal microbes: an axis of functional diversity among large marine consumers*", for your continued consideration as a Research Article in Proceedings of the Royal Society: Biological Sciences. On behalf of my co-authors, I would again like to thank you for the opportunity to further improve our manuscript.

Attached below you will find a detailed response to the reviewer's comments.

Sincerely on behalf of all co-authors,

Jarrod J Scott
Bocas del Toro, Panamá

Referee: 1

The authors made overall satisfactory changes to the manuscript, however I must maintain that the known ontogenetic diet shifts for these fish species should be contextualized in the manuscript. I understand the authors' point of view that they are looking at the general diet patterns commonly reported for these species, but the fact that there are known variations in diet preferences related to age (and perhaps even locality, Catano et al 2016) is also part of the state of the art. This knowledge is based on substantial research (to which the authors' themselves contributed), and importantly, their sampling DOES include fish from different life stages. Therefore, I also don't think omitting the original data on fish life-stage is correct. If the data were imprecise for surgeonfish please correct them (e.g. with fish size).

Research in this field is growing fast and in the future other researchers might want to use these data to tackle potentially more subtle effects of these reported dietary shifts. This could be easily achieved by using a GLMM and a random age/life-stage effect. Therefore, if the authors do not wish to test this age effect in the present manuscript, I strongly suggest the original metadata is kept and that the authors provide that background in Discussion.

The 'original metadata' referenced by the reviewer is, and has always been, provided in the supplementary material (Table S3). In that document we provide the following metadata for all individuals collected in this study:

- Weight (g)
- Total length (cm)
- Foregut length (cm)
- Midgut length (cm)
- Hindgut length (cm)
- Total gut length (cm)

These same data are also linked on the project website, itself referenced several times in the manuscript and SOM. These same data are also available in both the figshare (referenced in the SOM) and Dryad repos for the paper, and as part of the metadata in the ENA archive of the raw sequence data (referenced in the SOM). See below for a complete list.

In the original submission we included a **Life phase** category with the above-mentioned metadata for each individual. As we noted in our first revision, surgeonfish are not sequential hermaphrodites (like parrotfish) and so our "life phase" designation for the surgeonfish were selected based on the size of each animal—*large* and *small*. We felt that this language for surgeonfish was arbitrary and imprecise and thus chose to remove this classification (for surgeonfish only) from Table S3 during our re-submission. That said, the total length and weight of each individual (among other metadata) remained in the table, and the full content of the original metadata (Table S3) was part of the revised submission. In other words, aside from this minor improvement, no other element of the original metadata was altered in the revision.

It is very important to us that **all** of our data is easily accessible and our methods and results are transparent. We apologize if we failed to make our changes to the initial revision clear. The fish metadata referred to by the reviewer can be found in the following locations:

- Supplementary Table S3
- <https://doi.org/10.6084/m9.figshare.7379597>
- https://projectdigest.github.io/supplemental_material.html
- <https://www.ebi.ac.uk/ena/data/view/PRJEB28397>
- <https://istmobiome.github.io/DIGEST/Supplementary.html>

We hope the reviewer agrees that the original metadata is accessible.

While the reviewer is correct that our sampling included fish of different life stages and sizes, we would like to reiterate that our sample sizes for the microbial analyses are simply not large enough to look at ontogenetic shifts in gut microbiomes. We agree that understanding how the gut microbiome changes with ontogeny and under different ecological contexts would be a fruitful avenue for future research, and we have added a sentence to the Discussion section highlighting this (Lines 377-380):

Herbivorous fishes often display ontogenetic shifts in diet [28,42,45] as well as significant flexibility in foraging behavior among locations [32,45]. Targeted studies that link these ecological shifts to changes in their gut microbiomes should be a priority for future studies.

In addition, we made several small editorial changes that are included in the *track changes* version of the manuscript. We changed the order of Tables S1 and S2 to reflect their order in the text. These are the only changes we made to this version of the manuscript.